# Role of CD4/CD8 ratio on the incidence of tuberculosis in HIV-infected patients on antiretroviral therapy followed up for more than a decade

**Dawit Wolday**[1]*, **Yazezew Kebede**[1], **Dorsisa Legesse**[2], **Dawd S. Siraj**[3], **Joseph A. McBride**[3], **Mitchell J. Kirsch**[3], **Robert Striker**[3]

**1** Mekelle University College of Health Sciences, Mekelle, Ethiopia, **2** Hayat General Hospital, Addis Ababa, Ethiopia, **3** Division of Infectious Diseases, Department of Medicine, University of Wisconsin, Madison, Wisconsin, United States of America

* dawwol@gmail.com

## Abstract

### Background

The role of CD4/CD8 ratio on the incidence of tuberculosis (TB) in patients on antiretroviral therapy (ART) is unknown. Thus, we sought to determine whether the CD4/CD8 ratio was associated with development of TB in a cohort of HIV infected individuals on ART followed up for more than a decade in the setting of sub-Saharan Africa (SSA).

### Methods

The cohort comprised adult patients who started ART between 2001 and 2007 and followed for up to 15 years. Clinical data were collected in retrospective manner. Patients with an AIDS defining illness or a CD4 count <200 cell/μL were started with a combination of ART. The participants have clinic visits every 6 months and/or as needed. Poisson regression models were used to identify factors associated with development of incident TB. Kaplan-Meier curves were used to estimate the probability of incident TB while on ART.

### Results

A total of 347 patients with a median duration of follow-up on ART of 11.5 (IQR: 10.0–12.5) years were included. Incident TB developed in 47 patients during the 3259 person-years of follow-up, the majority (76.6%) occurred within five year of ART initiation. On univariate analysis, poor ART adherence (RR:2.57, 95% CI: 1.28–5.17), time-updated CD4 cell count of lower than 200 (RR: 4.86, 95%CI 2.33–10.15), or CD4 cell count between 200 and 500 (RR: 4.68, 95% CI: 2.17–10.09), time-updated CD8 cell count lower than 500 (RR: 2.83 95% CI 1.31–6.10), or CD8 cell count over 1000 (RR: 2.23, 95% CI: 1.12–4.45), time-updated CD4/CD8 ratio of less than 0.30 (RR: 6.00, 95% CI: 2.96–12.14), lack of normalization of CD4 T-cell count (RR: 6.13, 95% CI: 2.20–17.07), and virological failure (RR: 2.35 (95% CI: 1.17–4.71) were all associated with increased risk of incident TB. In multivariate

**Data Availability Statement:** Relevant data are available in the Figshare repository at: https://doi.org/10.6084/m9.figshare.12251330.

**Funding:** The study was supported by the Division of Infectious Diseases, Department of Medicine, University of Wisconsin, Madison, USA.

**Competing interests:** The authors have declared that no competing interests exist.

analysis, however, time-updated CD4/CD8 ratio of less than 0.30 (adjusted RR: 4.08, 95% CI: 1.31–12.68) was the only factor associated with increased risk of developing incident TB (p = 0.015). Similar results were obtained in a sensitivity analysis by including only those virally suppressed patients (n = 233, 69% of all patients). In this group, CD4/CD8 ratio of less than 0.30 was associated with development of incident TB (adjusted RR: 4.02, 95% CI: 1.14–14.19, p = 0.031). Overall, the incidence rate of TB in patients with an updated CD4/CD8 ratio of less than 0.30 was more than 5-fold higher when compared with those with a ratio more than 0.45.

## Conclusion

Low CD4/CD8 ratio is independently associated with an increased risk of incident TB despite viral suppression. CD4/CD8 ratio may serve as a biomarker for identifying patients at risk of TB in patients on ART in the setting of SSA.

## Introduction

Cumulative findings have demonstrated that antiretroviral therapy (ART) reduces the risk of developing tuberculosis (TB) significantly [1–18]. Despite the advent of ART, however, TB remains the major cause of death and morbidity among patients co-infected with HIV-1 in sub-Saharan Africa (SSA) [19–22]. The mechanism(s) related to the development of TB in patients taking ART, however, remains to be elucidated.

Effective ART suppresses HIV replication allowing progressive CD4+ T-cell recovery that is usually accompanied by persistent expansion of CD8+ T-cells. As a consequence, there is an incomplete restoration of the CD4/CD8 T-cell ratio, which very often remains inverted [23–26]. While it is well established that HIV, and specifically low CD4 cells in patients with HIV increase the susceptibility to TB [27], this strong relationship becomes obscured once patients start ART and their CD4 count rises. While CD4 counts rise fairly fast on HIV medications, the CD4/CD8 ratio does not rise or at best rises quite slowly and new evidence suggests this may be a marker for continuing HIV related immune dysfunction even when on ART [28].

We hypothesized that HIV positive patients on ART with low CD4/CD8 ratios are more likely to be diagnosed with incident TB than HIV positive patients with higher ratios. The CD4/CD8 ratio may serve as a marker of chronic immune activation, and may be clinically useful in identifying HIV-1 infected patients at risk for TB. Therefore, we sought to determine whether the CD4/CD8 ratio was associated with development of TB in a cohort of HIV infected individuals on ART followed up for more than a decade in the setting of SSA.

## Methods

### Study population

This retrospective cohort was enrolled at Hayat General Hospital's ART Clinic in Addis Ababa, Ethiopia, which delivers HIV treatment and care. The cohort consists of adult patients who started ART between 2001 and 2007 and have been followed until 2017. Patients with an AIDS defining illness or a CD4 count <200 cell/μL were started on with a combination of ART, based on National ART Guidelines [29]. First-line ART comprised of zidovudine (AZT) or stavudine (d4T) or tenofovir (TDF) in combination with lamivudine (3TC) plus non-nucleoside reverse transcriptase inhibitors, nevirapine (NVP) or efavirenz (EFV). All patients were

offered isoniazide preventive therapy (IPT) for preventing TB as of 2010. The participants had clinic visits every 6 months and/or as needed for monitoring treatment outcomes. Laboratory tests performed during each visit included complete blood cell count, clinical chemistry, urine analysis, and immune-profiles (as measured by CD4 cell count, CD8 cell count and CD4/CD8 ratio). HIV-1 viral load was done for patients after suspecting virological failure. The diagnosis of TB was based on the National Tuberculosis and Leprosy Control Guidelines of Ethiopia [30]. Accordingly, TB was diagnosed based on either one or a combination of the following: sputum microscopy, culture, chest radiography, fine-needle aspiration for lymphadenopathy, histopathology, or high index of clinical suspicion followed by response to treatment.

## Definitions

Patients presenting with TB at time or within one month of ART initiation were considered as prevalent TB. The development of TB following one month of ART initiation until censoring the study (death, loss to follow-up, transfer-out, or termination of the study) was considered as incident TB.

CD4+ T-cell count is considered normalized if above 500 cells (lower limit value of healthy adult Ethiopians [31, 32]), and CD4/CD8 ratio was considered normalized if above 1.0 [33, 34]. Viral suppression was defined as the attainment of HIV-1 plasma viral load of <400 copies/mL on two subsequent measurements. Poor adherence was defined as intake of prescribed medications <95%, based on National ART Guidelines [29].

## Statistical analysis

Baseline characteristics for continuous variables were expressed as the median with interquartile range (IQR), and for categorical variables as proportions. Chi-square test was used to compare categorical variables and Wilcoxon rank-sum test for not normally distributed variables.

The overall incidence rate of TB was calculated per 100 person years at risk. Patients were censored at the time of incident TB event or at the time of death, transfer to other clinic, loss to follow-up, or the last visit date before December 2017. Risk factors associated with the development of incident TB were analyzed using Poisson regression analysis. We opted to use Poisson regression analysis based on the small frequency of TB events in the cohort [1].

We first analyzed the association between the development of incident TB and baseline demographic (age, gender), clinical (HIV disease stage) characteristics, and baseline laboratory biomarkers including CD4 cell count, CD8 cell count, CD4/CD8 ratio and viral load in univariate analysis. The CD4 cell counts were built in three categories (<200, between 200 and 500, and >500 cells/μL), the CD8 cell counts in three categories (<500, between 500 and 1000, and >1000 cells/μL), and the CD4/CD8 ratio in three categories (<0.30, between 0.30 and 0.45, and >0.45). The cut-offs were chosen based on previous reports showing association of the biomarkers with AIDS-related events [33, 34].

In addition, time-dependent variables including ART adherence, toxicity, ART switching as well as updated CD4 cell count, CD8 cell count, CD4/CD8 ratio, normalized CD4 cell count (>500 cells), normalized CD4/CD8 ratio (≥1.0), and plasma viral load were included in the initial bivariate analysis. Then a multivariate analysis of predictors of incident TB was done including all variables that were significant in univariate analysis. Sensitivity analysis was done to examine the effect of viral suppression by excluding those who failed to attain viral suppression (<400 copies/mL) following initiation of ART.

Kaplan-Meier curves were used to estimate the probability of incident TB in patients on ART across explanatory variables and statistical differences were tested using log-rank test. P

values <0.05 were considered statistically significant. Data was entered on excel and exported for analysis by STATA (Statistical package v. 14.0, StataCorp, Texas, USA).

## Ethical considerations

The study was reviewed and approved by the Addis Ababa City Administration Health Bureau Research Ethics Review Committee, Addis Ababa, Ethiopia and the Institutional Review Board (IRB) of the University of Wisconsin, Madison, USA.

# Results

## Characteristics of study participants

Socio-demographic and baseline clinical features are shown in Table 1. A total of 347 (of which 36.3% were female) fulfilling recruitment criteria were included in the study. The median age at ART initiation was 40 (IQR: 33–46) years, the majority (61.4%) being Stage III and IV WHO clinical stage. We compared baseline characteristics between patients who developed TB and those who remained free from incident TB. We found that, there was no statistically significant difference in all of the characteristics between the two groups.

The CD4 cell count increased from baseline median of 127 cells (IQR: 60–197) to 411 cells (IQR: 225–602) after a median of 11.5 (IQR: 10.0–12.5) years of follow up on ART (p<0.0001). Overall, only 36.3% of the cohort has normalized CD4 cell count (>500 cells) at censoring the study. CD8 cell counts remained somewhat stable through-out treatment. In contrast CD4/CD8 ratio trends revealed continuous increase throughout the course of ART, and increased from a median baseline of 0.13 (IQR: 0.08–0.22) to a median of 0.49 (IQR: 0.26–0.72, at the time of censoring the study (p<0.001). In addition, only 9.7% of the cohort has normalized ratio (≥ 1.0) at censoring the study.

## TB incidence rates

Incident TB developed in 47 patients during the 3259 person-years of follow-up, accounting for an incidence rate of 1.44 per 100 person-year follow-up (95% CI: 1.08–1.92). The majority were extra-pulmonary tuberculosis (72%). The median time to developing TB was 1.5 years (IQR: 0.5 to 5.0). The median age of individuals who developed incident TB was 38 (IQR: 31–50) years. Of the 47 incident TB cases, the majority (76.6%) occurred within five year of ART initiation, accounting for an incidence rate of 53.03 per 100 person-year follow-up (95% CI: 38.25–73.51). On the contrary, the remaining incident TB cases occurred within 10 years and 15 years of follow-up, resulting in the incidence rate of 1.63 (95% CI 0.82–3.27) and 0.11 (95% CI 0.04–0.34) per 100 person-years of follow-up, respectively [S1 Table]. The incidence rate of TB in patients with an updated CD4/CD8 ratio of more than 0.45 was 0.47 per 100 person-year follow-up (95% CI: 0.25–0.87) and those with ratio between 0.30 and 0.45 was 0.85 per 100 person-year follow-up (95% CI: 0.35–2.05). However, when the ratio was less than 0.30, the incidence rate of TB was significantly higher (6.055 per 100 person-years of follow-up, 95% CI 4.28–8.55; p<0.0001) when compared with those with a ratio more than 0.45 (Fig 1).

We did further sensitivity analysis by excluding those who failed to achieve viral suppression. In this group, incident TB developed in 28 patients during the 2492 person-years of follow-up, accounting for an incidence rate of 1.12 per 100 person-year follow-up (95% CI: 0.78–1.63). The incidence rate of TB in this virally suppressed group with an updated CD4/CD8 ratio of more than 0.45 was 0.46 per 100 person-year follow-up (95% CI: 0.23–0.92) and those with ratio between 0.30 and 0.45 was 0.87 per 100 person-year follow-up (95% CI: 0.33–2.31). On the contrary, the incidence rate of TB was significantly higher (5.41 per 100 person-years

**Table 1. Baseline characteristics of study participants at initiation of ART.**

| Variable | | Overall (n = 347) | No incident TB (n = 300) | Incident TB (n = 47) | p-value |
|---|---|---|---|---|---|
| **Age (median years, IQR)** | | 40 (33–46) | 40 (33–46) | 38 (31–50) | 0.992 |
| **Gender** | **Male** | 221 (63.7) | 189 (63.0) | 32 (68.1) | |
| | **Female** | 126 (36.3) | 111 (37.0) | 15 (31.9) | 0.500 |
| **WHO Clinical Stage** | **I & II** | 134 (38.6) | 119 (39.7) | 15 (31.9) | |
| | **III & IV** | 213 (61.4) | 181 (60.3) | 32 (68.1) | 0.310 |
| **CD4 cell count (cells/µL)[a]** | **Median (IQR)** | 127 (60–197) | 132 (60–197) | 103 (68–190) | 0.521 |
| | **<200** | 263 (76.7) | 228 (76.8) | 35 (76.1) | |
| | **200–500** | 78 (22.7) | 67 (22.5) | 11 (23.9) | 0.842 |
| | **>500** | 2 (0.6) | 2 (0.7) | 0 (0.0) | |
| **CD8 cell count (cells/µL)[b]** | **Median (IQR)** | 891 (603–1300) | 905 (624–1279) | 775 (532–1419) | 0.323 |
| | **<500** | 54 (17.1) | 44 (16.2) | 10 (22.7) | |
| | **500–1000** | 128 (40.5) | 110 (40.4) | 18 (40.9) | 0.498 |
| | **>1000** | 134 (42.4) | 118 (43.4) | 16 (36.4) | |
| **CD4/CD8 ratio[b]** | **Median (IQR)** | 0.13 (0.08–0.22) | 0.13 (0.08–0.22) | 0.14 (0.09–0.21) | 0.853 |
| | **<0.30** | 277 (87.7) | 238 (87.5) | 39 (88.6) | |
| | **0.30–0.45** | 23 (7.3) | 20 (7.4) | 3 (6.8) | 0.977 |
| | **>0.45** | 16 (5.1) | 14 (5.1) | 2 (4.6) | |
| **HIV RNA (log₁₀ copies/mL)[c]** | **Median (IQR)** | 5.10 (4.55–5.46) | 5.11 (4.36–5.45) | 4.91 (4.80–5.67) | 0.629 |
| **Follow-up (years)** | **Median (IQR)** | 11.5 (10.0–12.5) | 11.5 (10.0–13.0) | 10.5 (8.0–12.0) | 0.090 |
| **ART regimen** | **AZT+3TC+ EFV** | 123 (35.5) | 109 (36.3) | 14 (29.8) | |
| | **d4T+3TC+ EFV** | 82 (23.6) | 71 (23.7) | 11 (23.4) | |
| | **TDF+3TC+EFV** | 51 (14.7) | 43 (14.3) | 8 (17.0) | |
| | **d4T+3TC+ NVP** | 50 (14.4) | 41 (13.7) | 9 (19.2) | |
| | **AZT+3TC+ NVP** | 25 (7.2) | 22 (7.3) | 3 (6.4) | |
| | **TDF+3TC+NVP** | 5 (1.4) | 5 (1.7) | 0 (0.0) | |
| | **PI containing** | 7 (2.0) | 6 (2.0) | 1 (2.2) | |
| | **Other** | 4 (1.2) | 3 (1.0) | 1 (2.2) | 0.523 |
| **HIV disease progression events** | **Other ADEs** | 33 (9.5) | 33 (11.0) | 0 (0.0) | 0.018 |
| | **NADEs** | 67 (19.3) | 62 (20.6) | 5 (10.9) | 0.120 |
| | **Death** | 54 (15.6) | 44 (14.7) | 10 (21.3) | 0.245 |

Data are numbers (%) unless otherwise stated. ART, antiretroviral therapy; TB, tuberculosis; IQR, interquartile range; ADE, AIDS-defining events; NADE, none AIDS-defining events.

[a]Data missing for 4 patients.

[b]Data missing for 31 patients.

[c]Data only for 86 patients.

of follow-up, 95% CI 3.32–8.83; p<0.001) in those with a ratio less than 0.30 compared to those with a ratio more than 0.45. In addition, patients with a ratio less than 0.30 and with viral failure did not exhibit significantly higher TB incidence (5.05 per 100 person-years of follow-up, 95% CI 2.52–10.09) [S2 Table]. The results suggest that viral suppression did not alter significantly the incidence rate of TB.

CD4 cell count, CD8 cell count, CD4/CD8 ratio and viral load at time of TB diagnosis in patients on ART who developed incident TB to those who remained free from TB is shown in Fig 2. The median baseline CD4 cell count among those who developed TB was not significantly different than that of patients who remained free from TB (p = 0.336). In addition, the median CD4 count at the time of TB diagnosis was significantly higher than at baseline (213

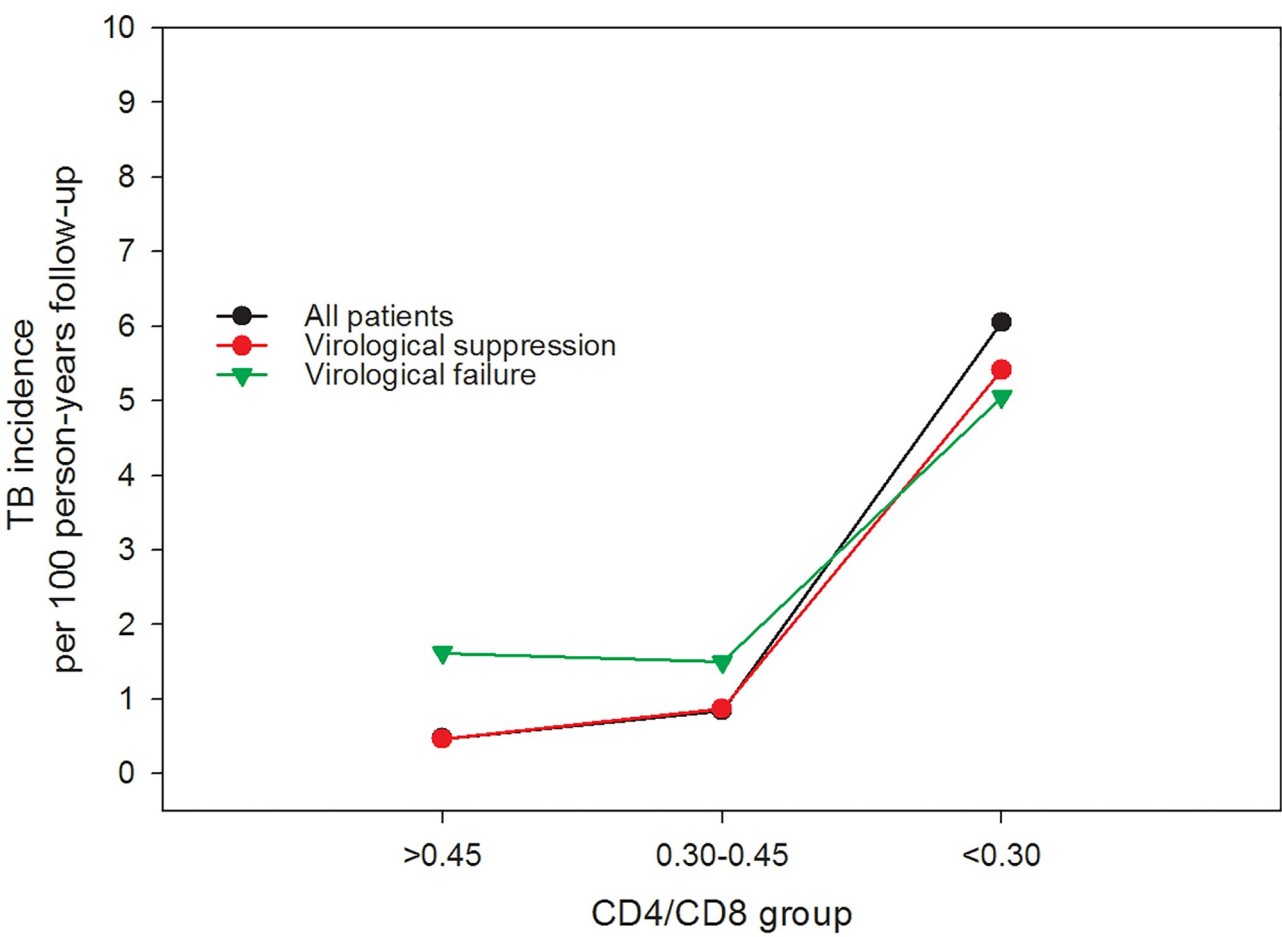

**Fig 1. The incidence rate of tuberculosis (TB) in the different CD4/CD8 ratio group.**

cells/µL; IQR: 76–325 vs. 103 cells/µL; IQR: 68–190, respectively (p = 0.002). Likewise, among the patients who remained free from TB, the time-updated CD4 cell count increased significantly compared to baseline (450 cells/µL; IQR: 280–626 vs. 132 cells/µL; IQR: 60–197, respectively (p<0.0001). However, among the patients who developed incident TB, the median CD4 cell count at the time of TB diagnosis (213 cells/µL; IQR: 76–325) was significantly lower than the time-updated cell count (450 cells/µL; IQR: 280–626) in those who remained free from TB (p<0.0001). The CD4/CD8 ratio increased significantly in both patient group irrespective of developing TB (Fig 2C). Nevertheless, despite similar baseline CD4/CD8 ratio in the patients who developed TB compared to those who remained free from TB (0.14; IQR: 0.09–0.21 vs. 0.13; IQR: 0.08–0.22, respectively; p = 0.612), the median CD4/CD8 ratio at the time of TB diagnosis was significantly lower than the time-updated ratio of those who remained free from TB (0.17; IQR: 0.11–0.37 vs. 0.52; IQR: 0.31–0.78, respectively; p<0.0001). Both CD8 cell count (Fig 2B) and viral load levels (Fig 2D) in patients who developed TB were not different than the values in those who remained free from TB.

The probability of TB-free survival in the cohort with an updated CD4/CD8 ratio of more than 0.45 was significantly higher than in the patients with a ratio less than 0.30, including only those with viral suppression (Fig 3). In addition, a similar pattern was noted among those who exhibited virological failure [S1 Fig].

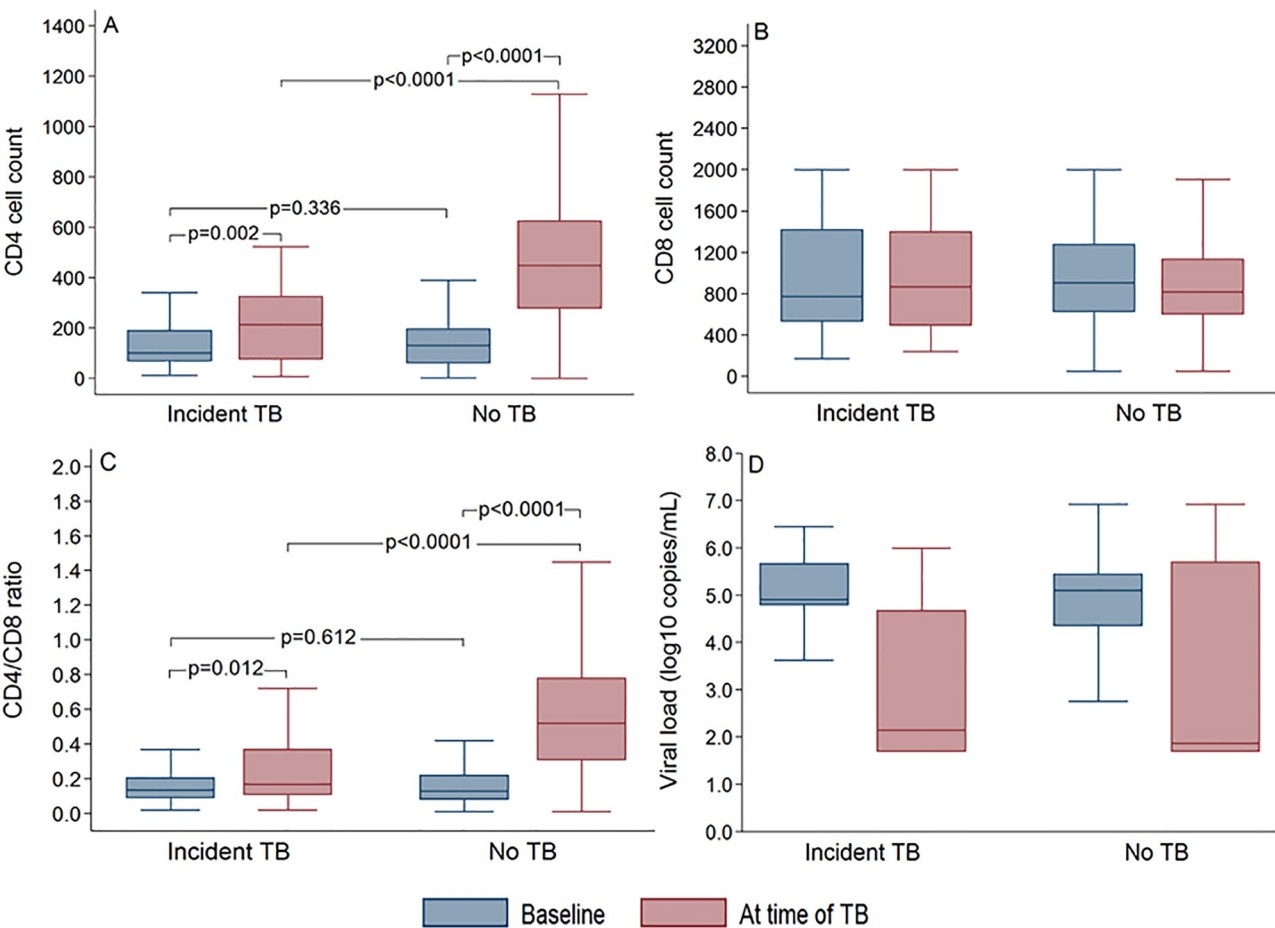

**Fig 2. CD4 cell count (A), CD8 cell count (B), CD4/CD8 ratio (C) and viral load levels (D) at time of TB in patients who developed incident TB vs. those who remained free from TB.** Data are presented as medians (IQR).

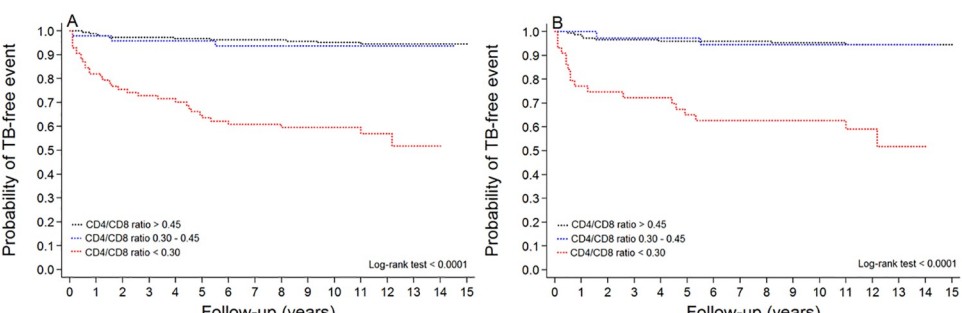

**Fig 3. Kaplan-Meier estimates for incident TB in patients on antiretroviral therapy.** Time-updated CD4/CD8 ratio in all cases (A), and in those with virological suppression.

## Risk factors associated with incident TB

On the univariate analysis (Table 2), poor ART adherence (RR:2.57, 95% CI: 1.28–5.17), time-updated CD4 cell count of lower than 200 (RR: 4.86, 95%CI 2.33–10.15), or CD4 cell count between 200 and 500 (RR: 4.68, 95%CI: 2.17–10.09), time-updated CD8 cell count lower than 500 (RR: 2.83 95% CI 1.31–6.10), time-updated CD8 cell count over 1000 (RR: 2.23, 95% CI: 1.12–4.45), time-updated CD4/CD8 ratio of less than 0.30 (RR: 6.00, 95% CI: 2.96–12.14), lack

**Table 2. Relative risk of tuberculosis in HIV-infected patients on antiretroviral treatment[a].**

| Variable | Unadjusted Relative Risk (95% CI) | P-value | Adjusted Relative Risk (95% CI) | P-value |
|---|---|---|---|---|
| **Age (years)** | | | | |
| < 50 | 1 | - - | - - | - - |
| ≥ 50 | 1.51 (0.68–3.75) | 0.313 | - - | - - |
| **Gender** | | | - - | - - |
| Male | 1 | - - | - - | - - |
| Female | 0.82 (0.45–1.52) | 0.531 | - - | - - |
| **WHO stage III and IV (*vs* I and II)** | 1.34 (0.73–2.48) | 0.347 | - - | - - |
| **Adverse drug events** | 1.42 (0.64–3.18) | 0.389 | - - | - - |
| **Poor adherence** | 2.57 (1.28–5.17) | <0.0001 | 1.08 (0.45–2.61) | 0.865 |
| **ART not switched** | 0.72 (0.39–1.31) | 0.284 | - - | - - |
| **Baseline CD4+ T-cell count** | | | | |
| > 500 | 1 | - - | - - | - |
| 200–500 | 0.84 (0.46–1.51) | 0.549 | - - | - - |
| < 200 | 1.06 (0.73–1.46) | 0.860 | - - | - - |
| **Baseline CD8+ T-cell count** | | | | |
| 500–1000 | 1 | - - | | |
| >1000 | 0.85 (0.43–1.67) | 0.634 | - - | - |
| <500 | 1.32 (0.61–2.85) | 0.485 | - - | - - |
| **Baseline CD4/CD8 ratio** | | | | |
| > 0.45 | 1 | - - | - - | - |
| 0.30–0.45 | 1.04 (0.17–6.24) | 0.963 | - - | - - |
| < 0.30 | 1.12 (0.27–4.66) | 0.870 | - - | - - |
| **Time-updated CD4+ T-cell count** | | | | |
| > 500 | 1 | - - | 1 | - |
| 200–500 | 4.68 (2.17–10.09) | <0.0001 | 1.91 (0.70–5.23) | 0.206 |
| < 200 | 4.86 (2.33–10.15) | <0.0001 | 1.44 (0.41–5.05) | 0.574 |
| **Time-updated CD8+ T-cell count** | | | | |
| 500–1000 | 1 | - - | 1 | - - |
| >1000 | 2.23 (1.12–4.45) | 0.023 | 1.22 (0.51–2.93) | 0.662 |
| <500 | 2.83 (1.31–6.10) | 0.008 | 2.29 (0.97–5.40) | 0.059 |
| **Time-updated CD4/CD8 ratio** | | | | |
| > 0.45 | 1 | - - | 1 | - - |
| 0.30–0.45 | 1.17 (0.32–4.25) | 0.813 | 0.94 (0.24–3.70) | 0.930 |
| < 0.30 | 6.00 (2.96–12.14) | <0.0001 | 4.08 (1.31–12.68) | 0.015 |
| **CD4+ T-cell count not normalized** | 6.13 (2.20–17.07) | 0.001 | 1.76 (0.47–6.51) | 0.399 |
| **CD4/CD8 ratio not normalized** | 2.93 (0.71–12.08) | 0.137 | - - | - - |
| **Virological failure[b]** | 2.35 (1.17–4.71) | 0.017 | 1.06 (0.46–2.48) | 0.887 |

[a]All cases (n = 347) with and without viral suppression included in model

[b]Sequential viral load data available for 272 (78.4%) patients

of normalization of CD4 T-cell count (RR: 6.13, 95% CI: 2.20–17.07), and viral failure (RR: 2.35 (95% CI: 1.17–4.71) were all associated with increased risk of developing TB. On the other hand age, gender, WHO stage, adverse drug events and not switching ART were not associated with risk of TB. In multivariate analysis, however, time-updated CD4/CD8 ratio of less than 0.30 (adj. RR: 4.08, 95% CI: 1.31–12.68) was the only factor associated with increased risk of developing TB (p = 0.015).

In sensitivity analysis, we only included those virally suppressed patients (n = 233, 69% of all patients). The results are shown in Table 3. On univariate analysis, time-updated CD4 cell

**Table 3. Relative risk of tuberculosis in HIV-infected patients on antiretroviral treatment[a].**

| Variable | Unadjusted Relative Risk (95% CI) | *P*-value | Adjusted Relative Risk (95% CI) | *P*-value |
|---|---|---|---|---|
| **Age (years)** | | | | |
| < 50 | 1 | - - | - - | - - |
| ≥ 50 | 1.98 (0.75–5.22) | 0.165 | - - | - - |
| **Gender** | | | - - | - - |
| Male | 1 | - - | - - | - - |
| Female | 0.84 (0.38–1.86) | 0.667 | - - | - - |
| **WHO stage III and IV (*vs* I and II)** | 2.14 (0.91–5.03) | 0.081 | - - | - - |
| **Adverse drug events** | 1.08 (0.32–3.65) | 0.897 | - - | - - |
| **Poor adherence** | 2.60 (0.99–6.83) | 0.053 | - - | - - |
| **ART not switched** | 0.66 (0.29–1.49) | 0.319 | - - | - - |
| **Baseline CD4+ T-cell count** | | | | |
| > 500 | 1 | - - | - - | - |
| 200–500 | 0.84 (0.46–1.51) | 0.549 | - - | - - |
| < 200 | 1.06 (0.73–1.46) | 0.860 | - - | - - |
| **Baseline CD8+ T-cell count** | | | | |
| 500–1000 | 1 | - - | | |
| >1000 | 0.43 (0.15–1.24) | 0.117 | - - | - |
| <500 | 1.55 (0.54–4.45) | 0.420 | - - | - - |
| **Baseline CD4/CD8 ratio** | | | | |
| > 0.45 | 1 | - - | - - | - |
| 0.30–0.45 | 2.04 (0.17–6.24) | 0.963 | - - | - - |
| < 0.30 | 1.10 (0.27–4.55) | 0.898 | - - | - - |
| **Time-updated CD4+ T-cell count** | | | | |
| > 500 | 1 | - - | 1 | - |
| 200–500 | 5.99 (2.21–16.18) | <0.0001 | 2.81 (0.73–10.72) | 0.132 |
| < 200 | 12.22 (4.52–33.05) | <0.0001 | 2.57 (0.50–13.18) | 0.258 |
| **Time-updated CD8+ T-cell count** | | | | |
| 500–1000 | 1 | - - | 1 | - - |
| >1000 | 1.22 (0.51–2.95) | 0.455 | 1.28 (0.48–3.44) | 0.625 |
| <500 | 2.84 (1.14–7.06) | 0.025 | 2.38 (0.90–6.26) | 0.080 |
| **Time-updated CD4/CD8 ratio** | | | | |
| > 0.45 | 1 | - - | 1 | - - |
| 0.30–0.45 | 1.03 (0.22–4.84) | 0.973 | 0.89 (0.17–4.56) | 0.886 |
| < 0.30 | 7.77 (3.38–17.86) | <0.0001 | 4.02 (1.14–14.19) | 0.031 |
| **CD4+ T-cell count not normalized** | 6.49 (1.96–21.49) | 0.002 | 1.55 (0.31–7.90) | 0.597 |
| **CD4/CD8 ratio not normalized** | 3.99 (0.54–29.36) | 0.174 | - - | - - |

[a]Only (n = 233, 67%) patients with viral suppression included in the model

count of lower than 200 (RR: 12.22, 95%CI 4.52–33.05), CD4 cell count between 200 and 500 (RR: 5.99, 95% CI: 2.21–16.18), time-updated CD8 cell count lower than 500 (RR: 2.84, 95% CI: 1.14–7.06), time-updated CD4/CD8 ratio of less than 0.30 (RR: 7.77, 95%CI: 3.38–17.86), and lack of normalization of CD4 T-cell count (RR: 6.49, 95% CI: 1.96–21.49) were associated with increased risk of developing TB. However, multivariate analysis revealed that time-updated CD4/CD8 ratio of less than 0.30 (adj. RR: 4.02, 95% CI: 1.14–14.19) was the only independent risk factor associated with development of TB (p = 0.031).

## Discussion

The overall TB incidence rate in our study is 1.44 (95% CI: 1.08–1.92) per 100 person years of follow-up. A more recent meta-analysis from Ethiopia reported a pooled TB incidence rate of 3.79 (95% CI: 2.03–5.55) per 100 person years based on 7 studies in patients who received ART and IPT [35]. Several reports have shown the complementary effects of ART and IPT in preventing the development of TB [15, 17, 36, 37]. All our cohort participants received IPT and this could also contribute to the lower incidence of TB. Overall, our result demonstrated about 2.6-fold lower incidence of TB than the previous studies reported from Ethiopia. However, longer treatment duration in our cohort might have resulted in a lower TB incidence rate, which is consistent with previous studies undertaken elsewhere in SSA [4–7]. Another important consideration is the fact that the cohort was followed in a private health-care facility. Thus, better management of TB in the private sector might have contributed to the findings of lower incidence rate of TB, as has been reported from India [17].

We noted that CD4 count at initiation of ART was not associated with risk of developing TB. This is consistent with the findings that ART significantly reduces the risk of TB across all CD4 count strata [7, 20]. We have shown previously that low CD4+ T-cell count precedes the development of TB in a cohort of HIV-positive Ethiopians before the advent of ART [27]. Likewise, in the current study, we found that the median CD4 cell count at the time of TB diagnosis was significantly lower than the counts in those who remained free from TB. Overall, these findings suggest that incident TB developed in the patients whose immunological responses to ART were suboptimal [38, 39].

Previous reports related to viral suppression and risk of developing TB produced conflicting reports. Some authors reported an increased risk of TB in patients on ART who did not achieve viral suppression [40], whereas others have demonstrated that achieving viral suppression is not associated with risk of TB development [3, 5, 41]. In our study, viral load levels did not differ significantly between those who developed TB and those who remained free of TB. In addition, we have demonstrated that there was no statistically significant difference on the incidence rate of TB in those who achieved viral suppression when compared with those who failed to attain viral suppression. Taken together, the findings suggest that, despite effective viral suppression after initiation of ART, CD4 recovery is suboptimal and eventually resulting in a residual risk of developing TB [38, 39]. Indeed, we and others have previously reported that incomplete restoration of *Mycobacterium tuberculosis*-specific CD4 T-cell responses despite ART [42, 43] contributes to increased incidence of TB.

Median CD4/CD8 ratio at time of ART initiation in patients who developed incident TB in our cohort was similar to recent reports [18, 44]. However, no significant difference in the average change of ratio over time on ART was seen in patients developing TB compared to patients remaining free of TB [44]. The investigators did not analyze the role of time-updated CD4/CD8 ratio on the incidence of TB. In our cohort, however, CD4/CD8 ratio at the time of TB diagnosis was significantly lower than those who remained free from TB. In our cohort followed-up for more than a decade, only few patients (≈10%) had normalized their CD4/CD8

ratio despite long-term viral suppression. Indeed, low CD4/CD8 ratio was strongly associated with risk of development of incident TB. Particularly, the incidence rate of TB when the CD4/CD8 ratio was less than 0.30 was more than 4-fold higher than that when the ratio was more than 0.45.

*M. tuberculosis*-mediated immunosuppression is a known phenomenon observed among TB patients resulting in low CD4 cell counts in both HIV-negative and HIV-positive patients [45–47]. The fact that our patients who developed TB had significant increase in CD4/CD8 ratio at the time of TB diagnosis compared to the ratio at enrolment suggests that the patients indeed responded immunologically to ART. Nonetheless, the increases in CD4/CD8 ratio among the patients who developed TB was significantly lower and suboptimal when compared to those who did not develop TB, resulting in their increased risk of developing TB. To the best of our knowledge, this is the first report demonstrating the impact of time-updated CD4/CD8 ratio on the incidence of TB in patients on ART. The strong association between a low CD4/CD8 ratio and the risk of TB might reflect the fact that a low ratio is a marker of a chronic immune activation. This suggestion is consistent with our previous studies conducted in Ethiopia whereby non-HIV infected adults exhibit CD4/CD8 inversion (<1.0) coupled with higher frequencies of activated effector memory cells [32, 48].

Strengths of our study include long duration of follow-up more than a decade, availability of immune profile data comprising of time-dependent change in CD4 cell count, CD8 cell count and CD4/CD8 ratio, and also the availability of viral load determinations for a large proportion (≈78%) of our cohort. Nonetheless, limitations related to our study include that our findings cannot be generalized to the entire population as the data is generated from the private sector and included only one study site. The diagnosis of TB in SSA setting is challenging as it is not definitive being without microbiological confirmation for all patients [14,15,21,47,49,50], as is for our cohort. All our patients received IPT for preventing TB as of 2010. IPT adherence data, however, were not available—which is another limitation of the study. Furthermore, the retrospective design of the study has limitations in respect to data completeness.

In conclusion, long-term ART confers a significant risk reduction in the development of TB, and in combination with IPT, it may contribute more to TB control in low-income countries. The fact that TB risk is independently associated with time-dependent CD4/CD8 ratio suggest that monitoring ratio levels may serve as a biomarker risk for TB among patients receiving ART.

## Supporting information

**S1 Table. Incidence of tuberculosis in HIV-infected patients on antiretroviral treatment followed for up to 15 years.**
(DOCX)

**S2 Table. Incidence of TB vs. time-updated CD4/CD8 category.**
(DOCX)

**S1 Fig. Kaplan-Meier estimates for incident TB in patients on antiretroviral therapy with updated CD4/CD8 ratio category.** Data are from those with virological failure.
(TIF)

## Acknowledgments

We thank all patients for participating in the study. We would like to thank the management and staff of Hayat General Hospital, College of Health Sciences, Addis Ababa, Ethiopia for their technical support and allowing us to conduct this study.

## Author Contributions

**Conceptualization:** Dawit Wolday, Dawd S. Siraj, Robert Striker.

**Data curation:** Dawit Wolday, Dorsisa Legesse, Mitchell J. Kirsch, Robert Striker.

**Formal analysis:** Dawit Wolday, Yazezew Kebede, Dawd S. Siraj, Joseph A. McBride, Mitchell J. Kirsch, Robert Striker.

**Funding acquisition:** Dawit Wolday, Dawd S. Siraj, Robert Striker.

**Investigation:** Dawit Wolday, Yazezew Kebede, Dawd S. Siraj, Joseph A. McBride, Mitchell J. Kirsch, Robert Striker.

**Methodology:** Dawit Wolday, Dawd S. Siraj, Robert Striker.

**Project administration:** Dawit Wolday, Dorsisa Legesse, Robert Striker.

**Resources:** Robert Striker.

**Supervision:** Dawit Wolday, Dorsisa Legesse, Robert Striker.

**Validation:** Dawit Wolday, Robert Striker.

**Writing – original draft:** Dawit Wolday, Yazezew Kebede.

**Writing – review & editing:** Dawit Wolday, Dawd S. Siraj, Joseph A. McBride, Robert Striker.

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
