## [Decision Letter · Decision Letter 0]

18 Feb 2020

PONE-D-19-29695

Role of CD4/CD8 ratio on the incidence of tuberculosis in HIV-infected patients on antiretroviral therapy followed up for more than a decade

PLOS ONE

Dear Wolday

Thank you for submitting your manuscript to PLOS ONE. After careful consideration, we feel that it has merit but does not fully meet PLOS ONE’s publication criteria as it currently stands. Therefore, we invite you to submit a revised version of the manuscript that addresses the points raised during the review process.

The longitudinal aspect of the study is its strength, yet the analysis presented in the manuscript is essentially a cross-sectional analysis, so misses the opportunity to better understand if low CD4/CD8 ratio is an early predictor of future TB risk, vs instead just a reflection of Mtb-mediated immunosupression around time of presentation with active TB.  Probably the answer is a mix of both, but the results as presented miss an important opportunity. The authors need to address this critical point. 

We would appreciate receiving your revised manuscript by Apr 03 2020 11:59PM. To enhance the reproducibility of your results, we recommend that if applicable you deposit your laboratory protocols in protocols.io, where a protocol can be assigned its own identifier (DOI) such that it can be cited independently in the future. For instructions see: http://journals.plos.org/plosone/s/submission-guidelines#loc-laboratory-protocols

We look forward to receiving your revised manuscript.

Kind regards,

Alan Landay

Academic Editor

PLOS ONE

Journal Requirements:

1. We note that you have indicated that data from this study are available upon request. PLOS only allows data to be available upon request if there are legal or ethical restrictions on sharing data publicly. For information on unacceptable data access restrictions, please see http://journals.plos.org/plosone/s/data-availability#loc-unacceptable-data-access-restrictions.

Reviewers' comments:

Reviewer's Responses to Questions

**Comments to the Author**

1. Is the manuscript technically sound, and do the data support the conclusions?

Reviewer #1: Yes

Reviewer #2: Partly

2. Has the statistical analysis been performed appropriately and rigorously? 

Reviewer #1: Yes

Reviewer #2: Yes

3. Have the authors made all data underlying the findings in their manuscript fully available?

Reviewer #1: Yes

Reviewer #2: No

4. Is the manuscript presented in an intelligible fashion and written in standard English?

Reviewer #1: Yes

Reviewer #2: Yes

5. Review Comments to the Author

Reviewer #1: There have been several studies reported on the possibility of using CD4/CD8 ratio as a clinical tool to evaluate patients' immune status.

It has already been well established that a persistently lower CD4/CD8 ratio during otherwise effective antiretroviral therapy is associated with increased innate and adaptive immune activation and immunosenescent phenotype, thereby leading to increased risk of morbidity and mortality.

The study though being retrospective, and has followed up a cohort of HIV infected individuals for more than a decade, suffers from several limitations such as the inconsistency in the method of TB diagnosis which is very critical for this study's objective and the literature review could have been wider.

Reviewer #2: This manuscript describes a retrospective study of adults who started ART between 2001 and 2007 and were followed for a median of 11.5 years, including periodic assessments of CD4 & CD8. The underlying hypothesis was that PLWH on ART with low CD4/CD8 ratios were more likely to be diagnosed with incident TB than their counterparts with higher CD4/CD8. Key findings included a low overall TB incidence rate (1.44/100 py) and that CD4/CD8 < 0.3 at the time of TB diagnosis was significantly associated with incident TB in multivariate analysis.

Major Comments

1. As written, the manuscript focuses on immunologic profiles at the time of incident TB diagnosis-- thus the analysis appears to mainly be cross-sectional and not include the longitudinal data. The longitudinal data are a real strength of the study, and are essential for understanding whether persistently low CD4/CD8 over time is a risk factor (and perhaps marker for) future TB. This is a missed opportunity.

2. Isoniazid preventive therapy is not included as a variable in analyses of TB risk, yet has been shown by other groups to strongly influence development of future TB. The Methods section states that all participants were offered IPT and the Discussion section states that all patients received IPT. Nevertheless, it would be unusual for IPT uptake or adherence to be 100%. If uptake/adherence data are available, their inclusion in analyses of TB risk is recommended. If uptake/adherence data are not available, then this should be so indicated in the Discussion and presented as a potential weakness.

3. The Discussion, lines 331-333, states that there was “no difference in TB incidence rate between those who achieved viral suppression vs. those who did not.” However, data are provided only for the overall group and the group who achieved viral suppression. Please either provide data and statistics for the comparison of those who achieved vs did not achieve viral suppression, or revise the Discussion sentence.

Minor Comments

1. The Discussion mentions that data collection was retrospective. This important design feature should be clearly indicated in the Abstact and in the Methods section.

6. PLOS authors have the option to publish the peer review history of their article (what does this mean?). If published, this will include your full peer review and any attached files.

Reviewer #1: No

Reviewer #2: No

---

## [Author Response · Author response to Decision Letter 0]

13 Mar 2020

We thank the academic editor and reviewers for their critical evaluation and their constructive criticisms. 

We provide the following response to the comments raised:

Academic editor:

• We agree with the comment that we used CD4/CD8 ratio data available around the time of incident TB. We have now included an additional analysis (see revised figure 2) showing longitudinal data in the cohort who developed incident TB vs. those who did not develop TB. The results suggest that though patients who developed TB respond to ART, the response is indeed smaller and suboptimal as compared to the responses noted among the patients who did not develop TB. 

Major comments:

Reviewer #1:

• The diagnosis of TB was based on the national guidelines. We have added a reference for this in the revised manuscript. The majority of the patients in our cohort indeed presented with extrapulmonary TB (72%), which is expected in HIV-infected patients. Diagnosis for such cases is confirmed by pathology. In addition, the challenge related to the diagnosis of TB in sub-Saharan Africa has been noted as a shortcoming of the study in the discussion section of the manuscript. We do have also included similar references from Ethiopia as well as other countries in Africa.

Reviewer #2:

1. We have added evidence from our longitudinal data showing that persistently low CD4/CD8 ratio overtime is a risk factor for developing incident TB (revised figure 2).

2. We agree with the reviewer that IPT impacts on development of TB. We have provided references for this in the original manuscript too. Our database lacks information with respect to IPT adherence. This shortcoming is now noted in the discussion section of the revised manuscript. 

3. Data for those who failed to achieve viral suppression has also been included (Figure 1 and Figure3).

Minor comment:

• That the data collected was retrospective has been noted both in the abstract and the methods sections.

---

## [Decision Letter · Decision Letter 1]

28 Apr 2020

Role of CD4/CD8 ratio on the incidence of tuberculosis in HIV-infected patients on antiretroviral therapy followed up for more than a decade

PONE-D-19-29695R1

Dear Dr. Wolday

We are pleased to inform you that your manuscript has been judged scientifically suitable for publication and will be formally accepted for publication once it complies with all outstanding technical requirements.

With kind regards,

Alan Landay

Academic Editor

PLOS ONE

Additional Editor Comments (optional):

Reviewers' comments:

Reviewer's Responses to Questions

**Comments to the Author**

1. If the authors have adequately addressed your comments raised in a previous round of review and you feel that this manuscript is now acceptable for publication, you may indicate that here to bypass the “Comments to the Author” section, enter your conflict of interest statement in the “Confidential to Editor” section, and submit your "Accept" recommendation.

Reviewer #1: All comments have been addressed

Reviewer #3: All comments have been addressed

2. Is the manuscript technically sound, and do the data support the conclusions?

Reviewer #1: Yes

Reviewer #3: Yes

3. Has the statistical analysis been performed appropriately and rigorously? 

Reviewer #1: Yes

Reviewer #3: Yes

4. Have the authors made all data underlying the findings in their manuscript fully available?

Reviewer #1: Yes

Reviewer #3: Yes

5. Is the manuscript presented in an intelligible fashion and written in standard English?

Reviewer #1: Yes

Reviewer #3: Yes

6. Review Comments to the Author

Reviewer #1: The comments have been appropriately addressed by the authors and the manuscript has been revised accordingly.

Reviewer #3: We appreciate your time responding to the comments from the initial review. The revised figures enhance the papers presentation.

7. PLOS authors have the option to publish the peer review history of their article (what does this mean?). If published, this will include your full peer review and any attached files.

Reviewer #1: No

Reviewer #3: No

---

## [Editor Report · Acceptance letter]

13 May 2020

PONE-D-19-29695R1 

Role of CD4/CD8 ratio on the incidence of tuberculosis in HIV-infected patients on antiretroviral therapy followed up for more than a decade 

Dear Dr. Wolday:

I am pleased to inform you that your manuscript has been deemed suitable for publication in PLOS ONE. Congratulations! Your manuscript is now with our production department. 

With kind regards,

on behalf of

Prof. Alan Landay 

Academic Editor

PLOS ONE